# Association between Dietary and Supplemental Antioxidants Intake and Lung Cancer Risk: Evidence from a Cancer Screening Trial

**DOI:** 10.3390/antiox12020338

**Published:** 2023-01-31

**Authors:** Jiaqi Yang, Sicheng Qian, Xiaona Na, Ai Zhao

**Affiliations:** 1Vanke School of Public Health, Tsinghua University, Beijing 100084, China; 2Bloomberg School of Public Health, Johns Hopkins University, Baltimore, MD 21205, USA; 3Faculty of Science, McGill University, Montreal, QC H3A 0G4, Canada

**Keywords:** antioxidant micronutrients, vitamins, minerals, lung cancer incidence, machine learning

## Abstract

Previous studies provided inconsistent results on the effects of antioxidant nutrient intake on lung cancer prevention. We aimed to evaluate the association between antioxidant consumption from food and supplemental sources and lung cancer incidence. Data were obtained from the Prostate, Lung, Colorectal, and Ovarian (PLCO) cancer screening trial. A total of 98,451 participants were included in the data analysis. We used a multivariable Cox proportional hazards regression model to calculate hazard ratios (HRs) and 95% confidence intervals (CIs) for the association between antioxidant intake and lung cancer risk. Dose-response assessments for individual nutrients were conducted. We also selected the model for the best combination of antioxidants for reducing lung cancer risk using machine learning methods. After the median follow-up of 12.2 years, 1642 new cases were identified. Intake of the calculated HRs indicated a trend for a higher quartile of food-based Composite Dietary Antioxidant Index (fCDAI) associated with a lower lung cancer risk after adjusting for covariates (HR_Q4vs.Q1_ = 0.64, 95% CI: 0.52, 0.79; *P* for trend < 0.001). Protective effects of dietary antioxidant intake were observed across all individual antioxidant micronutrients except magnesium. Random forests model suggested the dietary intake group of α-carotene, magnesium, vitamin C, vitamin E, lycopene, selenium, lutein, and zeaxanthin, and β-carotene had the most favorable effects on lung cancer prevention. Higher consumption of antioxidants from food sources has a protective effect against lung cancer, while no effects were shown in the supplemental group. It is recommended to consume a combination of various antioxidants due to the potential benefits from the interaction, while more research should be performed to investigate the underlying mechanisms of antioxidant synergic effects on lung cancer risk reduction.

## 1. Introduction

Lung cancer is one of the most diagnosed malignancies worldwide, with high incidence and mortality rates each year [1]. In 2020, there were 22.4 cases per 100,000 persons across all ages and both sexes, with an estimation of 2,206,771 new cases and 1,796,144 deaths worldwide [2]. Several risk factors for lung cancer development have been identified in previous studies, such as cigarette smoking, exposure to second-hand smoke, asbestos, and air pollution [1,3].

Evidence has also suggested that diet, such as antioxidant intake, is associated with lung cancer incidence [4,5]. Antioxidants, such as vitamins C, vitamin E, β-carotene, and other phytochemicals, have abilities to combat free radicals and inhibit oxidation [6]. Previous epidemiological studies have shown that higher consumption of antioxidant-rich fruits and vegetables is linked to certain cancer risk reduction, including lung cancer [6,7,8]. According to the current updates from World Cancer Research Fund (WCRF), there is moderate evidence indicating that the consumption of dietary sources of antioxidants, vegetables, and fruits may be a protective factor against lung cancer occurrence, as well as retinol- or carotenoids-rich foods [9]. However, the individual roles of antioxidant nutrients on lung cancer risk remain inconclusive. A well-documented antioxidant bioactive compound, β-carotene, has demonstrated an inverse association with lung cancer incidence in several studies [8,10]. Strong evidence has shown that high-dose β-carotene supplemental intake increases lung cancer risk among current and former smokers [9]. In addition, a systematic review and meta-analysis highlighted that the favorable effects of selenium supplementation have only been shown in populations with a lower baseline serum selenium status, while another meta-analysis noted an inverse association between selenium exposure and lung cancer risk without identifying a threshold effect in the dose-response analyses [11,12]. These studies yield mixed results on cancer development, which are sophisticated and require further research on the individual roles of antioxidants.

Many studies have examined the individual effects of certain antioxidant intake, while there are inadequate studies investigating the combined effects of antioxidant consumption from both dietary and supplemental sources. Due to the potential interactions between different antioxidants, separately looking at individual micronutrients could not fully account for the synergic effects of antioxidants intake on lung cancer risk. The food-based Composite Dietary Antioxidant Index (fCDAI) has been developed and used in multiple longitudinal studies to capture the overall antioxidant intake from various food groups and evaluate the associations with cancer risk [13,14,15]. We utilized this index to predict lung cancer risk in a generally healthy population. Meanwhile, we also employed machine learning methods to investigate the optimal antioxidant combination in lung cancer prevention.

Therefore, in this study, we aimed to examine the effects of the independent and combined intake of antioxidant nutrients on lung cancer risk, differentiate the effects based on dietary and supplemental sources, and investigate the optimal combination of individual antioxidants on lung cancer prevention in different subgroups in a large population-based cancer screening trial.

## 2. Materials and Methods

### 2.1. Study Population and Study Design

The data used in this study were from the Prostate, Lung, Colorectal, and Ovarian (PLCO) study, a cancer screening trial, which included a total of 155,000 participants recruited from 10 centers across the US. The trial aimed to evaluate the effects of screening exams on decreasing cancer mortality rates. Participants were enrolled first in 1993, randomized into the control arm (usual care) or the intervention arm (screening exams). Eligible participants were aged between 55 and 74 years old at the enrollment and free of prostate, lung, colorectal, or ovarian cancer history. Baseline information of participants was collected through a baseline questionnaire (BQ). Dietary history questionnaire (DHQ) has been provided to both arms since 1998. Around 77% or 113,000 participants completed DHQ with a 3-year median time into study collected. Subjects were followed for approximately 12 years to collect data on cancer diagnoses.

In total, there were 154,887 participants extracted from the PLCO trial. After excluding 53,155 individuals who failed to complete valid BQ and DHQ, 101,732 subjects remained in the study. Eligible participants who completed a valid BQ had no history of lung cancer prior to the trial but had time at risk for developing their first cancer. A valid DHQ was defined as the completion of the questionnaire with a completion date before date of death, missing no more than 7 frequency responses, and not having extreme energy intake (top and bottom 1% of each sex group). We further excluded 3281 participants due to the missing of important covariates (study arm, sex, education, body mass index, marital status, family history of lung cancer or any other cancer, smoking status, pack-years of cigarettes, and alcohol drinking status). Finally, there were 98,451 cases included in the analytical dataset. A flow diagram is presented in Figure 1.

The PLCO screening trial was approved by the Institutional Review Board of the National Cancer Institute (NCI), and written consent forms were obtained from subjects prior to and after the randomization for agreements of participating trial activities and screening. This Cancer Data Access System (CDAS) project was approved by NCI, and the project ID is PLCO-974.

### 2.2. Data Collection

Baseline data on demographics, medical history, and other risk factor information, such as smoking status, were self-reported by participants. In our study, we obtained age, sex, race, study arm, education, body mass index (BMI), marital status, family history of any cancer, family history of lung cancer, smoking status, pack-years of cigarettes, and alcohol drinking status. Dietary data were collected through DHQ, which included 156 questions to assess alcohol use, nutrient intake, supplement intake, total energy intake, and daily consumption of foods and beverages in grams and frequencies in the past year. Specifically, we obtained dietary intake of vitamin A, vitamin C, vitamin E, α-carotene, β-carotene, magnesium, selenium, zinc, lycopene, lutein, and zeaxanthin. Nutrition Data System for Research (NDS-R), along with U.S. Department of Agriculture (USDA)’s Continuing Survey of Food Intakes by Individuals (CSFII), were used to estimate the amount of nutrients intake. Additionally, daily supplemental intake of several vitamins and minerals, which was available, was also retrieved.

### 2.3. fCDAI Score Calculation

The food-based Composite Dietary Antioxidant Index (fCDAI) was used to calculate a summary score of individual’s dietary intake of antioxidants in respect of the mean intake of the entire cohort. The index has been validated in other studies, which found it to be inversely associated with several pro-inflammatory biomarkers and helpful in evaluating the effects of antioxidant intake on health outcomes [13,14,15]. An fCDAI score was calculated for each participant by estimating the dietary consumption of six antioxidants, including vitamins A, C, E, magnesium, selenium, and zinc, using the following formula:$$\mathrm{fCADI}=\sum_{i=1}^{6}\frac{xi-\mu i}{Si}$$
where *x*_i_ is the daily consumption of antioxidant *i*. μi stands for the mean intake of antioxidant *i* across entire study population, and S*_i_* is the SD for μi for antioxidant *i*.

### 2.4. Ascertment of Lung Cancer Cases

The outcome of interest was the incidence of lung cancer. Study participants self-reported lung cancer diagnoses through annual questionnaires. Abnormal chest x-ray screening, death certificates, and relative informed cases were followed up. The lung cancer diagnoses were all confirmed through medical record abstraction (MRA) later. The lung cancer histopathologic types derived from International Classification of Diseases for Oncology, 2nd Edition (ICD-O-2) morphology, including non-small cell lung cancer and small cell lung cancer. Carcinoid tumors were not considered during screening in the trial and were not included in this study as confirmed cases.

### 2.5. Statistical Analysis

We used Cox proportional hazards regression model to estimate the hazard ratio (HR) and its 95% confidence intervals (CIs) for the association between antioxidants intake and lung cancer risk. Time of follow-up time was defined as cohort entry date till the date of cancer diagnoses, death, drop-out, or the end of study through 2009. The daily antioxidant intake and fCDAI score of study population were divided into quartiles. Both dietary and supplemental intake were assessed for their associations with lung cancer incidence. Models were age-adjusted and additionally adjusted for other covariates, including study arm (usual care, screening), sex (male, female), education (<8 years, 8–11 years, high school, college, postgraduate), BMI (continuous), marital status (married, divorced, separated, widowed), family history of lung cancer (yes, no) or any other cancer (yes, no), smoking status (never, current, former), pack-years of cigarettes (continuous), and alcohol drinking status (yes, no). Tests for linear trend were conducted, and median intakes were used to denote corresponding quartiles in regression models. Dose-response analyses were further performed to examine the relationship between daily intake (dietary and supplemental) of antioxidants and lung cancer incidence using the restricted cubic splines. The 20th percentile of intake was set as the reference value. Lastly, we used random forest to select models of best combination of antioxidants for lung cancer prevention, grouped by sources of intake (food, supplements, all sources). All models adopted 80% of data as a train set and 20% of data as a test set. Average decrease in accuracy and mean-reduced Gini coefficient were two sorting methods used for random forests. All statistical analyses were performed by R Studio (4.2.1, Boston, MA, USA) and STATA (16.0, College Station, TX, USA).

## 3. Results

### 3.1. Baseline Characteristics

After the median follow-up of 12.2 years, there were 98,451 subjects, including 1642 cases of lung cancer being diagnosed. Table 1 shows the baseline characteristics of included participants by the quartile distribution of fCDAI scores. At the baseline, the mean (SD) age of subjects was 62.4 (5.28) years old. Males had a higher dietary antioxidant consumption than the female population (*p* < 0.001). Other significant differences were exhibited regarding age (*p* < 0.001), race (*p* < 0.001), study arm (*p* = 0.0023), education (*p* < 0.001), BMI (*p* < 0.001), marital status (*p* < 0.001), family history of any cancer (*p* = 0.0158), smoking status (*p* < 0.001), alcohol drinking status (*p* < 0.001), and total energy intake (*p* < 0.001). In addition, participants who consumed higher amounts of dietary antioxidants were more educated and more likely to be former cigarette smokers.

### 3.2. Association between Individual Antioxidants and Risk of Lung Cancer

Table 2 shows the associations between individual antioxidant nutrient intake from diets and supplements and lung cancer incidents among all subjects, stratified by quartiles of intake. α-carotene, β-carotene, vitamin A, vitamin C, vitamin E, magnesium, selenium, zinc, lycopene, lutein, and zeaxanthin were major antioxidants with high bioactivities we investigated. As shown in Table 2, both the age-adjusted and multi-adjusted models indicated that the higher dietary and total intake of β-carotene was associated with a decreased lung cancer risk. Compared with the lowest quartile, the highest quartile of total β-carotene intake, with a mean of 9775 mcg/day, appeared to be linked to a reduced lung cancer incidence (HR = 0.69, 95% CI: 0.59–0.80). A significant trend across quartiles was observed (p for trend < 0.001). A nonsignificant trend towards potentially harmful effects was shown among those who took daily β-carotene supplements (HR: 1.04, 95% CI: 0.94–1.15, *p* = 0.453) in the multi-adjusted model.

After adjusting multiple covariates, protective effects were also found in total and dietary intake of vitamin A, dietary vitamin C, total and dietary vitamin E, dietary selenium, dietary zinc, dietary lycopene, dietary lutein and zeaxanthin, and dietary α-carotene, where significant trends across quartiles were observed in each antioxidant nutrient. The lowest HR was observed in the third quartile of total and dietary vitamin C consumption, compared to HRs of other quartiles. When total vitamin C intake was treated as a continuous variable, a decreased and then slightly increased risk of lung cancer was observed using the restricted cubic spline model (reference value = 113.31 mg/day, p for nonlinear = 0.005) (Appendix A). A similar trend was observed in dietary lycopene (Appendix A). Significant nonlinear dose-response curves were also shown in total β-carotene, dietary α-carotene, dietary lutein, and zeaxanthin intake with a sharp decrease of risk at a lower dosage level followed by a continuous but slower decreased risk at higher dosage levels (Appendix A). Noted, there were no significant associations observed among supplemental β-carotene, vitamin A, vitamin C, vitamin E, magnesium, and zinc intake with lung cancer risk (Appendix A).

### 3.3. Association between fCADI Score and Risk of Lung Cancer

Table 3 demonstrates the associations between fCADI scores and lung cancer incidence by quartiles. The overall HRs showed that a higher quartile indicated a lower risk of lung cancer in the multi-adjusted model. Compared to the 1st quartile, the highest quartile of fCADI scores was linked to a 36% risk reduction in lung cancer (HR = 0.64, 95% CI: 0.52, 0.79). Lung cancer incidence decreased as fCADI scores increased (*P* for trend < 0.001).

Figure 2 and Table A1 show the results of the subgroup analyses in different populations. Pronounced protective linear trends were also observed among male, female, normal weight and overweight individuals, current cigarette smokers, current alcohol drinkers, and individuals with and without a family history of lung cancer. However, no effects were noted among individuals who have never smoked or formerly smoked and individuals who have never drunk or formerly drank.

### 3.4. Best Combination of Antioxidants Model Selection and Random Forest

Figure 3 shows the receiver operating characteristic (ROC) curve to illustrate the possibly optimal models of antioxidants combination on lung cancer incidence reduction, grouped by sources of intake. For the dietary intake group, the variable importance ranking was α-carotene, followed by magnesium, vitamin C, vitamin E, lycopene, selenium, lutein, and zeaxanthin, and β-carotene. When the mtry value (the random number used by each tree) equaled three, it had the lowest out-of-bag data error rate. The prediction accuracy was 0.88, and the area under the ROC curve (AUC) was 0.89, which indicated a good fit. The models for the supplemental group and total intake group were less acceptable than the dietary group, with AUC equaling 0.69 for both (Figure A1 and Figure A2).

## 4. Discussion

This study investigated the association between antioxidant micronutrient intake and the risk of lung cancer in a large US population, with the majority being Caucasians. After adjusting for age, sex, BMI, smoking, and other risk factors for lung cancer based on previous literature, we found an inverse association between overall dietary antioxidants intake and incidence of lung cancer, as illustrated by the fCADI score. Higher total individual antioxidant intakes, including β-carotene, vitamin A, and vitamin E, were also associated with a decreased risk of lung cancer. Protective effects of dietary antioxidant intake were observed across all individual antioxidant micronutrients except magnesium. However, dietary magnesium played the second most important role in the best dietary model selected by random forest for lung cancer prevention, and the antioxidant combination was α-carotene, magnesium, vitamin C, vitamin E, lycopene, selenium, lutein and zeaxanthin, and β-carotene. Moreover, we did not notice any significant effects of supplemental antioxidant intake on lung cancer risk. In subgroup analyses, protective effects were not modified by sex, BMI, or family history of lung cancer. Effect modifications were shown among never and former smokers and drinkers, and the advantages of total diet-derived antioxidants were only observed among current smokers and current alcohol drinkers.

Previous evidence suggested an inverse association between dietary antioxidant intake and several cancer risks, including prostate cancer, digestive cancer, and lung cancer [16,17,18]. In the present study, we confirmed previous results on lung cancer risk and moved the field forward by individually examining each micronutrient from both dietary and supplemental sources. We found that total β-carotene, vitamin A, and vitamin E intake was associated with a decreased risk of lung cancer, while we only observed protective effects from dietary vitamin C, selenium, and zinc. Moreover, we only obtained data on lycopene, lutein and zeaxanthin, and α-carotene from dietary sources, and protective effects were found in all of them. Magnesium was the only micronutrient not linked to lung cancer risk in both dietary and supplemental forms. The results from the present study were in line with another prospective population-based cohort study in the Netherlands, which demonstrated dietary zinc intake decreased the risk of lung cancer by 42%, while no association was observed with dietary magnesium [19].

Although the protective effects of the above nutrients were recorded in many previous studies and confirmed in the current study, one randomized controlled trial and meta-analyses did not advocate for the high-dose use of β-carotene, vitamin C, and vitamin E supplementation for cancer prevention purposes in the generally healthy population, as no beneficial effects on total cancer incidence and mortality were shown [20,21]. A recent Swedish study illustrated the potential biological basis behind the scenes [22]. Given that lung cancer cells require lots of energy, such as sugar, to grow and multiply rapidly, this faster energy-making process would cause tremendous oxidative stress on cells by generating free oxygen radicals [22,23]. The antioxidant supplements would support lung cancer cells to withstand the stress and thrive [22]. In line with the result of this present study that the daily use of β-carotene supplementation, with a mean of 295 mcg/day, might potentially have a 4% incremental risk of developing lung cancer. The effect was not significant, and this might be due to the dosage being relatively low compared to the dosage in clinical interventions, that we were unable to observe the potential harmful effects at a high dosage. In addition, since the baseline population was generally healthy, we deemed participants who reported the daily consumption of β-carotene supplementation did not take it separately but as part of the multivitamin formula, which might minimize the harmful effects of sole supplemental β-carotene. When β-carotene supplement is consumed individually, a high risk of cardiovascular outcomes is associated, and the harmful effects are more pronounced among current smokers [24]. This is probably due to the fact that under high oxygen concentrations, β-carotene undergoes a pro-oxidant mechanism, which may indirectly induce negative health consequences [24]. However, when β-carotene is obtained from food sources or consumed with other micronutrient supplements together, the interactions between different antioxidants may produce synergic effects on attenuating the pro-oxidant property of β-carotene under certain circumstances [24]. Besides β-carotene, the dose-response analyses in current study also revealed a decreased and slightly increased trend for total vitamin C intake, showing that moderate consumption (500 mg/day) was optimal for lung cancer prevention and the high intake of vitamin C should be treated with caution. Hence, the sources where the antioxidants being obtained from and the dosage level of antioxidants consumption yield mixed results on lung cancer incidence prevention, which requires more evidence to make firm conclusions in future studies.

It should be noticed that lifestyle factors also play an important role in effect modification. Previous large clinical trials have demonstrated that among smoking populations, β-carotene supplementation increases the risk of lung cancer, independent of tar or nicotine level of cigarettes, and this is due to the property of β-carotene as a pro-oxidant under the free-radical-rich condition of smokers [25,26,27,28]. In the present observational study, we noted a 4% nonsignificantly elevated risk among people who took β-carotene supplementation daily, while we did not observe a dose-response relationship between the supplementation and lung cancer events since a very small portion of participants consumed a high-dose basis (>30 mg/d). Additionally, in the current study, we noted that among current smokers and drinkers, a higher fCADI score was associated with a significant risk reduction of lung cancer. In the subgroup of current smokers, we detected a significant linear trend of protective effects of total diet-derived antioxidants intake, while the effects were not shown among never or former users. The results were consistent with a recent case-control study conducted in Canada that among heavy and moderate smokers, an inverse association was found between elevated dietary intakes of β-carotene, α-carotene, and lycopene in male and vitamin C in females and lung cancer risk [29]. However, current smokers only took up 9.3% of the total population in our study, and the finding should be treated with caution. Given the above, the results of the present study suggested that individuals with different lifestyles might have varied responses to the effects of antioxidants on lung cancer risk. Therefore, when conducting future dietary interventions, additional considerations should be made in assessing different lifestyle factors. More research is also needed to explore the interactions between antioxidant intake and lifestyle factors.

We applied the fCADI score to examine the combined effects of individual antioxidants and proved that jointly consumed antioxidants together could significantly reduce lung cancer risk. Furthermore, to the best of our knowledge, this is the first study using the machine learning method to select the optimal model of a random forest, rank the importance of antioxidants, and find the best combination by groups. Since previous studies yielded mixed results on the effects of diet-derived antioxidants in different cohort settings, we applied this method to incorporate more input parameters to fit the prediction model with a potentially higher accuracy level. We noticed that the dietary group model with antioxidants intake from food sources had the highest prediction value on lung cancer prevention, with α-carotene intake being the most important factor, while models for the other two groups did not present sufficient prediction value. The strong protective effects of α-carotene were also demonstrated in a pooled analysis of two large U.S. cohorts that increased α-carotene intake was associated with a 63% risk reduction of lung cancer among non-smokers [30]. Another population-based study using data from the Third Nutrition and Health Examination Survey (NHANES III) also showed that a high serum level of α-carotene at baseline was linked to a lower risk of lung cancer and a 46% risk reduction of death among current smokers [31]. Moreover, although we did not observe the protective effect of individual dietary intake of magnesium in this cohort, it was the second most important predictor in the model, after α-carotene. This might be due to the potential synergic effect which altered the association, while the underlying biological mechanisms are still unclear, which require further research in future studies.

Of note, the present study has several unique strengths. First, this study has a relatively long follow-up period, approximately 12 years, to collect lung cancer diagnoses. Second, the study cohort is very large and has a wide representativeness of the US population. Study participants were recruited across 10 screening centers in the US. Third, due to the nature of this prospective cohort study, the reverse causation bias could be reduced. Forth, in the subgroup analyses, we interestingly noted the inverse association between fCADI score and lung cancer incidence is more pronounced among current smokers and drinkers.

Nevertheless, several limitations need to be addressed. First, the dietary information was only collected once at the baseline, while the diets might change during the long follow-up period, which could lead to misclassifications. However, this could also help minimize the bias of reverse causation. Second, because DHQ was self-reported, the information might not be accurate enough, and recall bias could exist. Third, 36.4% of study participants were excluded due to failure to complete valid BQ or DHQ and missing important covariates. We could not evaluate the difference between the selected group and excluded group, leading to possible selection bias.

## 5. Conclusions

Our results suggested that a higher overall dietary antioxidant consumption from various food sources was associated with a lower risk of lung cancer. No protective effects were shown in antioxidant supplements. Lifestyle factors, including smoking and alcohol drinking, might modify the observed associations. In addition, future studies should investigate the potential interactions between vitamins and minerals in lung cancer prevention.

## Figures and Tables

**Figure 1 antioxidants-12-00338-f001:**
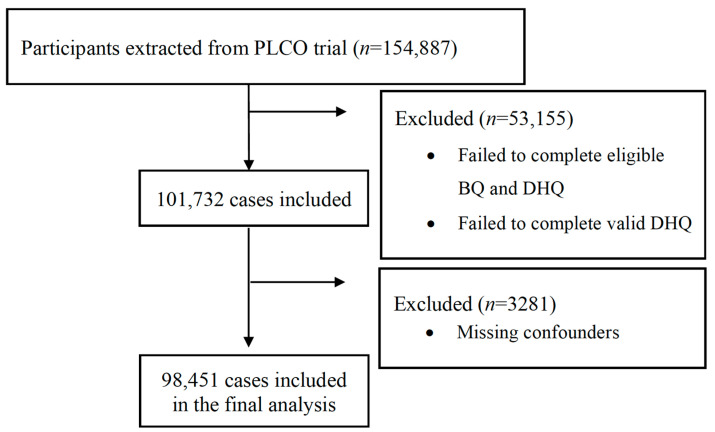
Flow diagram of selecting individuals from the PLCO trial.

**Figure 2 antioxidants-12-00338-f002:**
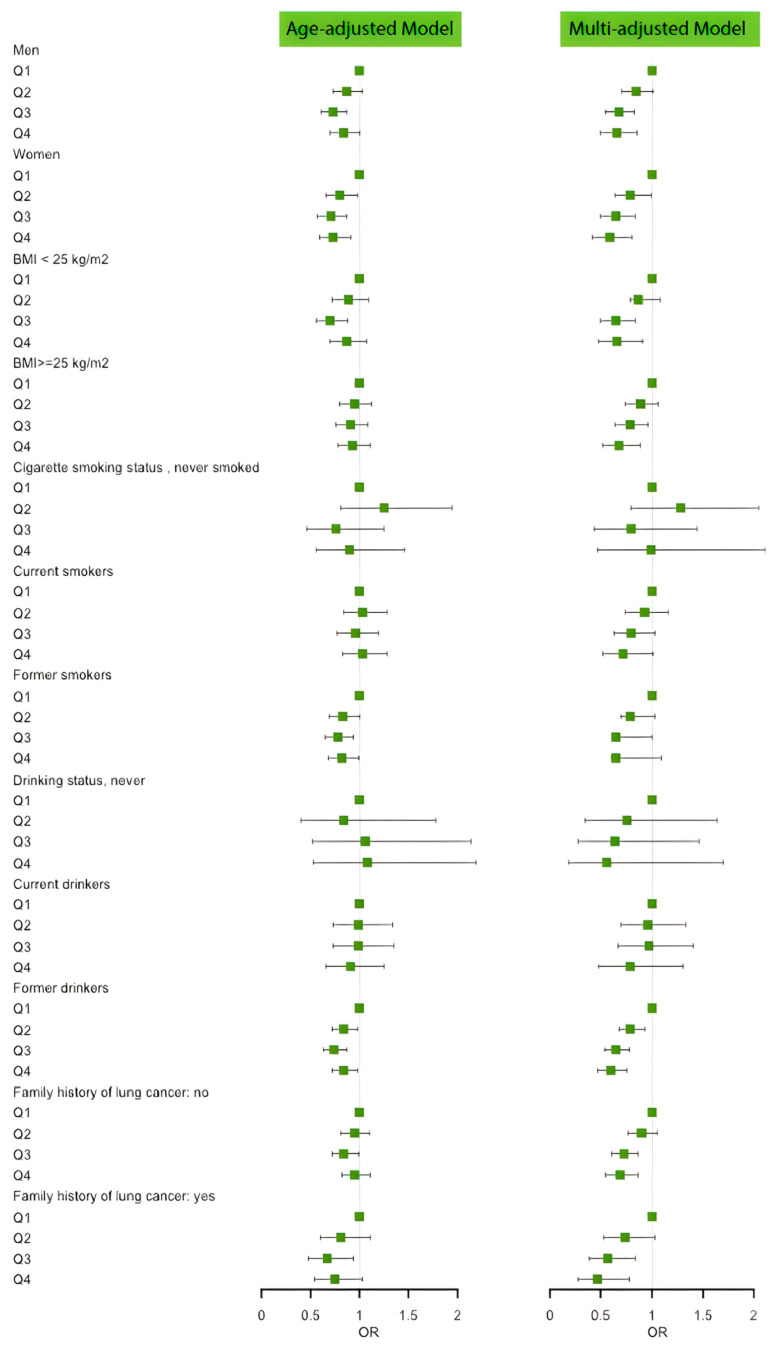
Forest plot of the associations between fCADI Score and lung cancer incidence by subgroups.

**Figure 3 antioxidants-12-00338-f003:**
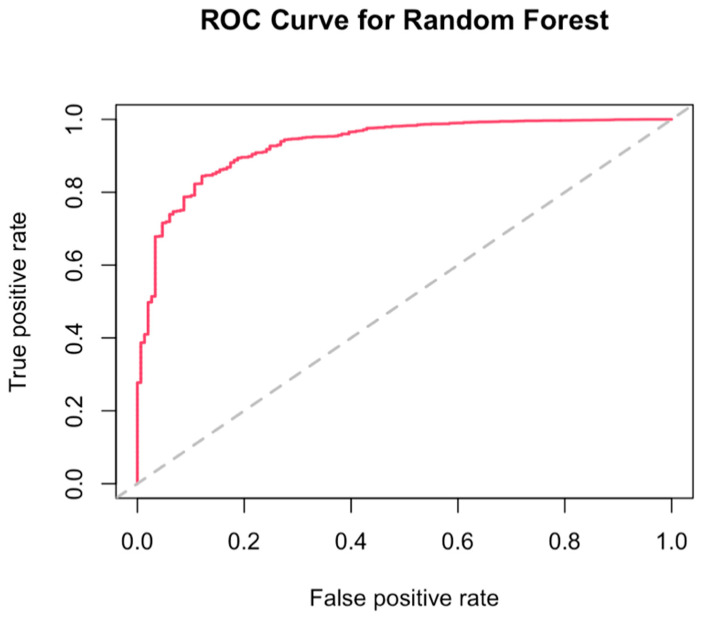
A receiver operating characteristic (ROC) curve for the dietary group.

**Table 1 antioxidants-12-00338-t001:** Baseline Characteristics of 98,451 participants from PLCO Cancer Screening Trial by fCDAI Quartiles.

Title 1	1 (*n* = 24,613)	2 (*n* = 24,613)	3 (*n* = 24,613)	4 (*n* = 24,612)	Overall (*n* = 98,451)	*p*-Value
Age						
Mean (SD)	62.7 (5.34)	62.5 (5.28)	62.4 (5.27)	62.0 (5.20)	62.4 (5.28)	<0.001
Median [Min, Max]	62.0 [53.0, 75.0]	62.0 [54.0, 74.0]	62.0 [54.0, 78.0]	61.0 [54.0, 75.0]	62.0 [53.0, 78.0]	
Sex						
Male	8861 (36.0%)	10,742 (43.6%)	12,585 (51.1%)	15,474 (62.9%)	47,662 (48.4%)	<0.001
Female	15,752 (64.0%)	13,871 (56.4%)	12,028 (48.9%)	9138 (37.1%)	50,789 (51.6%)	
Race						
White, Non-Hispanic	21,868 (88.8%)	22,605 (91.8%)	22,806 (92.7%)	22,299 (90.6%)	89,578 (91.0%)	<0.001
Black, Non-Hispanic	992 (4.0%)	730 (3.0%)	645 (2.6%)	844 (3.4%)	3211 (3.3%)	
Hispanic	394 (1.6%)	316 (1.3%)	344 (1.4%)	386 (1.6%)	1440 (1.5%)	
Asian	1182 (4.8%)	817 (3.3%)	664 (2.7%)	874 (3.6%)	3537 (3.6%)	
Pacific Islander	111 (0.5%)	103 (0.4%)	102 (0.4%)	139 (0.6%)	455 (0.5%)	
American Indian	58 (0.2%)	31 (0.1%)	45 (0.2%)	62 (0.3%)	196 (0.2%)	
Missing	8 (0.0%)	11 (0.0%)	7 (0.0%)	8 (0.0%)	34 (0.0%)	
Study arm						
Intervention	12,422 (50.5%)	12,468 (50.7%)	12,722 (51.7%)	12,795 (52.0%)	50,407 (51.2%)	0.00229
Control	12,191 (49.5%)	12,145 (49.3%)	11,891 (48.3%)	11,817 (48.0%)	48,044 (48.8%)	
Education						
Less Than 8 Years	177 (0.7%)	115 (0.5%)	138 (0.6%)	167 (0.7%)	597 (0.6%)	<0.001
8–11 Years	1539 (6.3%)	1307 (5.3%)	1203 (4.9%)	1216 (4.9%)	5265 (5.3%)	
12 Years or Completed High School	6938 (28.2%)	5883 (23.9%)	5284 (21.5%)	4815 (19.6%)	22,920 (23.3%)	
Post High School Training Other than College	3251 (13.2%)	3204 (13.0%)	3238 (13.2%)	3036 (12.3%)	12,729 (12.9%)	
Some College	5515 (22.4%)	5218 (21.2%)	5213 (21.2%)	5210 (21.2%)	21,156 (21.5%)	
College Graduate	3686 (15.0%)	4440 (18.0%)	4605 (18.7%)	4607 (18.7%)	17,338 (17.6%)	
Postgraduate	3507 (14.2%)	4446 (18.1%)	4932 (20.0%)	5561 (22.6%)	18,446 (18.7%)	
BMI						
0–18.5	186 (0.8%)	186 (0.8%)	156 (0.6%)	136 (0.6%)	664 (0.7%)	<0.001
18.5–25	8780 (35.7%)	8610 (35.0%)	8338 (33.9%)	7449 (30.3%)	33,177 (33.7%)	
25–30	10,041 (40.8%)	10,373 (42.1%)	10,595 (43.0%)	10,940 (44.4%)	41,949 (42.6%)	
30+	5606 (22.8%)	5444 (22.1%)	5524 (22.4%)	6087 (24.7%)	22,661 (23.0%)	
Marital status						
Married Or Living As Married	18,461 (75.0%)	19,464 (79.1%)	19,755 (80.3%)	19,575 (79.5%)	77,255 (78.5%)	<0.001
Widowed	2545 (10.3%)	2024 (8.2%)	1812 (7.4%)	1571 (6.4%)	7952 (8.1%)	
Divorced	2647 (10.8%)	2199 (8.9%)	2150 (8.7%)	2385 (9.7%)	9381 (9.5%)	
Separated	196 (0.8%)	162 (0.7%)	172 (0.7%)	228 (0.9%)	758 (0.8%)	
Never Married	764 (3.1%)	764 (3.1%)	724 (2.9%)	853 (3.5%)	3105 (3.2%)	
Family history of any cancer						
No	10,701 (43.5%)	10,811 (43.9%)	10,854 (44.1%)	11,075 (45.0%)	43,441 (44.1%)	0.0158
Yes, Immediate Family Member	2665 (10.8%)	2573 (10.5%)	2556 (10.4%)	2504 (10.2%)	10,298 (10.5%)	
Possibly Relative or Cancer Type Not Clear	632 (2.6%)	559 (2.3%)	581 (2.4%)	591 (2.4%)	2363 (2.4%)	
Smoking status						
Never Smoked Cigarettes	12,318 (50.0%)	12,152 (49.4%)	11,757 (47.8%)	11,310 (46.0%)	47,537 (48.3%)	<0.001
Current Cigarette Smoker	2579 (10.5%)	2204 (9.0%)	2135 (8.7%)	2189 (8.9%)	9107 (9.3%)	
Former Cigarette Smoker	9716 (39.5%)	10,257 (41.7%)	10,721 (43.6%)	11,113 (45.2%)	41,807 (42.5%)	
Alcohol drinking status						
Never	3008 (12.2%)	2593 (10.5%)	2236 (9.1%)	2103 (8.5%)	9940 (10.1%)	<0.001
Former	3759 (15.3%)	3494 (14.2%)	3446 (14.0%)	3570 (14.5%)	14,269 (14.5%)	
Current	17,001 (69.1%)	17,838 (72.5%)	18,323 (74.4%)	18,308 (74.4%)	71,470 (72.6%)	
Unknown	845 (3.4%)	688 (2.8%)	608 (2.5%)	631 (2.6%)	2772 (2.8%)	
Total energy						
Mean (SD)	1050 (300)	1480 (325)	1850 (411)	2570 (740)	1740 (735)	<0.001
Median [Min, Max]	1030 [466, 4970]	1450 [550, 5390]	1810 [622, 5540]	2450 [644, 5620]	1610 [466, 5620]	
fCADI						
Mean (SD)	−5.07 (1.23)	−2.07 (0.727)	0.682 (0.928)	6.46 (3.93)	−0.0000000675 (4.76)	<0.001
Median [Min, Max]	−4.85 [−10.1, −3.34]	−2.08 [−3.34, −0.805]	0.618 [−0.805, 2.43]	5.29 [2.43, 42.0]	−0.805 [−10.1, 42.0]	

**Table 2 antioxidants-12-00338-t002:** Hazard Ratios (HRs) of the association between total, dietary, and supplemental antioxidants intake and lung cancer incidence by quartiles of intake.

Nutrients	Range	Mean	Cohort	Cases	Incidence Rate (95% CI) ^a^		Age-Adjusted HR (95% CI), *p*-Value	Multi-Adjusted HR (95% CI), *p*-Value
Total β-carotene (mcg/day)								
Q1	<2278.77	1556.37	24,614	516	0.052 (0.047–0.056)		Reference group	Reference group
Q2	≥2278.77 to <3504.26	2864.17	24,616	455	0.045 (0.041–0.050)		0.86 (0.76–0.97), *p* = 0.017	0.98 (0.86–1.11), *p* = 0.761
Q3	≥3504.26 to <5665.90	4422.85	24,609	374	0.037 (0.034–0.041)		0.70 (0.61–0.80), *p* = 1.04 × 10^−7^	0.86 (0.75–0.99), *p* = 0.039
Q4	≥5665.90	9775.35	24,612	297	0.030 (0.027–0.033)		0.55 (0.48–0.63), *p* < 2 × 10^−16^	0.69 (0.59–0.80), *p* = 1.36 × 10^−6^
*P* for linear trend							P for trend < 2 × 10^−16^	P for trend = 1.75 × 10^−7^
Dietary β-carotene								
Q1	<1634.06	1137.74	24,613	547	0.055 (0.050–0.059)		Reference group	Reference group
Q2	≥1634.06 to <2654.56	2119.15	24,613	426	0.043 (0.039–0.047)		0.76 (0.67–0.86), *p* = 2.30 × 10^−5^	0.84 (0.74–0.95), *p* = 0.007
Q3	≥2654.56 to <4428.63	3423.88	24,612	379	0.038 (0.034–0.042)		0.67 (0.58- 0.76), *p* = 1.17 × ^10−9^	0.79 (0.69–0.91), *p* = 0.001
Q4	≥4428.63	7527.37	24,613	290	0.029 (0.026–0.033)		0.51 (0.44–0.59), *p* ≤ 2 × 10^−16^	0.62 (0.53–0.72), *p* = 1.03 × 10^−9^
*P* for linear trend							P for trend < 2 × 10^−16^	P for trend = 3.79 × 10^−9^
β-carotene supplements								
No	0	0	37,113	651	0.065 (0.060–0.070)		Reference group	Reference group
Yes	≤2700	294.93	61,338	991	0.099 (0.093–0.105)		0.93 (0.84–1.02), *p* = 0.125	1.04 (0.94–1.15), *p* = 0.453
Total vitamin A (IU/day)								
Q1	<7422.5	4991.567	24,613	490	0.049 (0.045–0.054)		Reference group	Reference group
Q2	≥7422.5 to <10,810.0	9139.481	24,616	475	0.048 (0.043–0.052)		0.94 (0.83–1.07), *p* = 0.378	1.05 (0.92–1.19), *p* = 0.450
Q3	≥10,810.0 to <15,489.5	12,873.09	24,609	375	0.037 (0.034–0.041)		0.73 (0.64–0.84), *p* = 4.58 × 10^−6^	0.86 (0.74–0.98), *p* = 0.029
Q4	≥15,489.5	22,986.43	24,613	302	0.030 (0.027–0.034)		0.58 (0.51–0.67), *p* = 2.23 × 10^−13^	0.72 (0.61–0.84), *p* = 2.28 × 10^−5^
*P* for linear trend							P for trend = 9.16 × 10^−16^	P for trend = 1.60 × 10^−6^
Dietary vitamin A								
Q1	<4993.5	3652.943	24,615	518	0.052 (0.048–0.056)		Reference group	Reference group
Q2	≥4993.5 to <7513.7	6212.687	24,612	446	0.045 (0.041–0.049)		0.83 (0.73–0.94), *p* = 0.004	0.85 (0.75–0.97), *p* = 0.0155
Q3	≥7513.7 to <11,610.5	9303.387	24,612	375	0.038 (0.034–0.041)		0.68 (0.60–0.78), *p* = 2.21 × 10^−8^	0.75 (0.65–0.87), *p* = 0.0001
Q4	≥11,610.5	18,619.62	24,612	303	0.030 (0.027–0.034)		0.55 (0.48–0.64), *p* < 2 × 10^−16^	0.63 (0.54–0.74), *p* = 1.73 × 10^−8^
*P* for linear trend							P for trend <2 × 10^−16^	P for trend = 2.88 × 10^−8^
Vitamin A supplements								
No	0	0	37,183	653	0.065 (0.060–0.071)		Reference group	Reference group
Yes	≤30,000	4901.844	61,268	989	0.099 (0.093–0.105)		0.93 (0.84–1.02), *p* = 0.124	1.03 (0.93–1.14), *p* = 0.578
Total vitamin C (mg/day)								
Q1	<127.85	85.368	24,613	536	0.054 (0.049–0.058)		Reference group	Reference group
Q2	≥127.85 to <210.22	165.918	24,615	364	0.036 (0.033–0.040)		0.65 (0.57–0.74), *p* = 1.40 × 10^−10^	0.82 (0.71–0.93), *p* = 0.003
Q3	≥210.22 to <563.31	328.331	24,610	358	0.036 (0.032–0.040)		0.64 (0.56–0.73), *p* = 3.44 × 10^−11^	0.81 (0.70–0.93), *p* = 0.003
Q4	≥563.31	925.296	24,613	384	0.038 (0.035–0.042)		0.67 (0.59–0.76), *p* = 2.03 × 10^−9^	0.82 (0.72–0.94), *p* = 0.005
P for linear trend							P for trend = 0.0001	P for trend = 0.0625
Dietary vitamin C								
Q1	<80.03	56.195	24,616	558	0.056 (0.051–0.061)		Reference group	Reference group
Q2	≥80.03 to <120.77	100.178	24,614	393	0.039 (0.036–0.043)		0.67 (0.59–0.76), *p* = 8.22 × 10^−10^	0.81 (0.71–0.92), *p* = 0.001
Q3	≥120.77 to <171.92	144.357	24,614	342	0.034 (0.031–0.038)		0.56 (0.49–0.64), *p* < 2 × 10^−16^	0.72 (0.62–0.82), *p* = 2.98 × 10^−6^
Q4	≥171.92	251.426	24,607	349	0.035 (0.031–0.039)		0.58 (0.50–0.66), *p* = 6.34 × 10^−16^	0.73 (0.63–0.85), *p* = 4.43 × 10^−5^
*P* for linear trend							P for trend = 2.84 × 10^−15^	P for trend = 4.04 × 10−5
Vitamin C supplements								
No	0	0	26,670	506	0.051 (0.046–0.055)		Reference group	Reference group
Yes	≤2102.86	326.698	71,781	1136	0.114 (0.107–0.120)		0.83 (0.75–0.93), *p* = 0.0007	0.98 (0.88–1.09), *p* = 0.656
Total vitamin E (mg/day)								
Q1	<13.23	7.452	24,614	465	0.046 (0.042–0.051)		Reference group	Reference group
Q2	≥13.23 to <34.42	24.914	24,613	452	0.045 (0.041–0.050)		0.97 (0.86–1.11), *p* = 0.685	1.07 (0.93–1.21), *p* = 0.348
Q3	≥34.42 to <291.94	189.001	24,612	342	0.034 (0.031–0.038)		0.71 (0.62–0.81), *p* = 1.23 × 10^−6^	0.83 (0.72–0.96), *p* = 0.012
Q4	≥291.94	387.723	24,612	383	0.038 (0.035–0.042)		0.79 (0.69–0.91), *p* = 0.0007	0.89 (0.78–1.02), *p* = 0.103
*P* for linear trend							*P* for trend = 5.81 × 10^−7^	*P* for trend = 0.0016
Dietary vitamin E								
Q1	<4.971	3.768	24,644	477	0.048 (0.044–0.052)		Reference group	Reference group
Q2	≥4.971 to <6.901	5.924	24,613	391	0.039 (0.035–0.043)		0.83 (0.73–0.95), *p* = 0.007	0.82 (0.71–0.92), *p* = 0.006
Q3	≥6.901 to <9.591	8.119	24,582	399	0.040 (0.036–0.044)		0.86 (0.75–0.98), *p* = 0.028	0.82 (0.62–0.82), *p* = 0.008
Q4	≥9.591	13.531	24,612	375	0.038 (0.034–0.041)		0.84 (0.73–0.96), *p* = 0.009	0.72 (0.63–0.85), *p* = 0.0005
*P* for linear trend							*P* for trend = 0.031	*P* for trend = 0.0017
Vitamin E supplements								
No	0	0	25,133	490	0.049 (0.045–0.054)		Reference group	Reference group
Yes	≤ 690.1	192.445	73,318	1152	0.115 (0.109–0.122)		0.79 (0.71–0.88), *p* = 1.17 × 10^−5^	0.91 (0.82–1.02), *p* = 0.097
Total magnesium (mg/day)								
Q1	<273.73	213.779	24,615	397	0.040 (0.036–0.044)		Reference group	Reference group
Q2	≥273.73 to <354.16	314.646	24,612	392	0.039 (0.036–0.043)		0.98 (0.86–1.13), *p* = 0.824	0.95 (0.82–1.10), *p* = 0.477
Q3	≥354.16 to <446.29	397.172	24,612	418	0.042 (0.038–0.046)		1.04 (0.91–1.20), *p* = 0.547	0.97 (0.83–1.12), *p* = 0.657
Q4	≥446.29	554.459	24,612	435	0.044 (0.040–0.048)		1.12 (0.97–1.28), *p* = 0.112	0.92 (0.77–1.10), *p* = 0.369
*P* for linear trend							*P* for trend = 0.0684	*P* for trend = 0.4266
Dietary magnesium								
Q1	<233.03	183.725	24,616	389	0.039 (0.035–0.043)		Reference group	Reference group
Q2	≥233.03 to <303.16	268.121	24,610	388	0.039 (0.035–0.043)		0.98 (0.85–1.13), *p* = 0.822	0.94 (0.81–1.08), *p* = 0.371
Q3	≥303.16 to <389.11	342.821	24,613	404	0.040 (0.037–0.045)		1.03 (0.89–1.18), *p* = 0.724	0.90 (0.77–1.05), *p* = 0.187
Q4	≥389.11	495.427	24,612	461	0.046 (0.042–0.051)		1.21 (1.05–1.38), *p* = 0.007	0.94 (0.77–1.14), *p* = 0.546
P for linear trend							*P* for trend = 0.0025	*P* for trend = 0.5619
Magnesium supplements								
No	0	0	45,192	806	0.081 (0.075–0.086)		Reference group	Reference group
Yes	≤100	87.787	53,259	836	0.084 (0.078–0.089)		0.89 (0.81–0.98), *p* = 0.021	1.00 (0.90–1.10), *p* = 0.933
Total selenium (mcg/day)								
Q1	<59.57	45.411	24,621	416	0.042 (0.038–0.046)		Reference group	Reference group
Q2	≥59.57 to <81.65	70.528	24,612	383	0.038 (0.035–0.042)		0.93 (0.81–1.07), *p* = 0.336	0.88 (0.76–1.02), *p* = 0.079
Q3	≥81.65 to <110.78	94.956	24,613	401	0.040 (0.036–0.044)		1.00 (0.87–1.15), *p* = 0.972	0.86 (0.73–1.00), *p* = 0.054
Q4	≥110.78	147.879	24,605	442	0.044 (0.040–0.049)		1.16 (1.01–1.33), *p* = 0.031	0.84 (0.68–1.03), *p* = 0.093
*P* for linear trend							*P* for trend = 0.0086	*P* for trend = 0.1306
Dietary selenium								
Q1	<58.33	44.648	24,624	416	0.042 (0.038–0.046)		Reference group	Reference group
Q2	≥58.33 to <79.27	68.767	24,611	379	0.038 (0.034–0.042)		0.92 (0.80–1.06), *p* = 0.263	0.86 (0.74–0.99), *p* = 0.037
Q3	≥79.27 to <106.76	91.811	24,605	414	0.041 (0.038–0.046)		1.04 (0.91–1.19), *p* = 0.588	0.88 (0.75–1.03), *p* = 0.105
Q4	≥106.76	142.739	24,611	433	0.043 (0.039–0.048)		1.14 (1.00–1.30), *p* = 0.059	0.75 (0.61–0.93), *p* = 0.010
*P* for linear trend							*P* for trend = 0.0139	*P* for trend = 0.0187
Selenium supplements								
No	0	0	92,249	1550	0.155 (0.147–0.163)		Reference group	Reference group
Yes	≤42.86	42.86	6202	92	0.009 (0.008–0.011)		0.88 (0.71–1.08), *p* = 0.219	0.89 (0.72–1.09), *p* = 0.263
Total zinc (mg/day)								
Q1	<9.95	7.012	24,616	446	0.045 (0.041–0.049)		Reference group	Reference group
Q2	≥9.95 to <19.84	14.354	24,632	397	0.040 (0.036–0.044)		0.93 (0.81–1.06), *p* = 0.265	0.89 (0.77–1.02), *p* = 0.098
Q3	≥19.84 to <25.77	22.811	24,602	395	0.040 (0.036–0.044)		0.90 (0.78–1.03), *p* = 0.113	0.94 (0.82–1.08), *p* = 0.399
Q4	≥25.77	34.401	24,601	404	0.040 (0.037–0.045)		0.92 (0.81–1.06), *p* = 0.243	0.86 (0.74–1.00), *p* = 0.053
*P* for linear trend							P for trend = 0.219	P for trend = 0.1519
Dietary zinc								
Q1	<7.00	5.381	24,626	413	0.041 (0.038–0.045)		Reference group	Reference group
Q2	≥7.00 to <9.48	8.234	24,688	398	0.040 (0.036–0.044)		0.97 (0.84–1.11), *p* = 0.650	0.94 (0.81–1.08), *p* = 0.367
Q3	≥9.48 to <12.90	11.016	24,528	406	0.041 (0.037–0.045)		1.02 (0.89–1.17), *p* = 0.812	0.93 (0.79–1.09), *p* = 0.351
Q4	≥12.90	17.816	24,609	425	0.042 (0.039–0.047)		1.09 (0.95–1.25), *p* = 0.204	0.82 (0.67–0.99), *p* = 0.041
*P* for linear trend							*P* for trend = 0.119	*P* for trend = 0.0430
Zinc supplements								
No	0	0	42,257	765	0.076 (0.071–0.082)		Reference group	Reference group
Yes	≤ 36.43	15.825	56,194	877	0.088 (0.082–0.094)		0.87 (0.79–0.96), *p* = 0.004	0.96 (0.87–1.06), *p* = 0.460
Dietary lycopene (mg/day)								
Q1	<3108.61	2170.982	24,613	469	0.047 (0.043–0.051)		Reference group	Reference group
Q2	≥3108.61 to <4761.09	3908.915	24,613	390	0.039 (0.035–0.043)		0.85 (0.75–0.98), *p* = 0.0220	0.87 (0.76–0.99), *p* = 0.0408
Q3	≥4761.09 to <7491.66	5948.377	24,612	373	0.037 (0.034–0.041)		0.85 (0.74–0.97), *p* = 0.0175	0.84 (0.73–0.97), *p* = 0.0157
Q4	≥7491.66	13,823.78	24,613	410	0.041 (0.037–0.045)		0.96 (0.84–1.10), *p* = 0.5347	0.83 (0.71–0.96), *p* = 0.0144
*P* for linear trend							*P* for trend = 0.977	*P* for trend = 0.0437
Dietary lutein and zeaxanthin (mg/day)								
Q1	<1246.48	897.2885	24,613	511	0.051 (0.047–0.056)		Reference group	Reference group
Q2	≥1246.48 to <1906.74	1563.793	24,613	428	0.043 (0.039–0.047)		0.83 (0.73–0.94), *p* = 0.0042	0.95 (0.83–1.08), *p* = 0.4021
Q3	≥1906.74 to <3002.04	2379.873	24,612	364	0.036 (0.033–0.040)		0.71 (0.62–0.81), *p* = 4.47 × 10^−7^	0.83 (0.72–0.96), *p* = 0.0112
Q4	≥3002.04	5674.224	24,613	339	0.034 (0.030–0.038)		0.66 (0.57–0.76), *p* = 2.84 × 10^−9^	0.78 (0.67–0.91), *p* = 0.0015
*P* for linear trend							*P* for trend = 1.02 × 10^−8^	*P* for trend = 0.0011
Dietary α-carotene (mg/day)								
Q1	<295.22	193.6037	24,615	532	0.053 (0.049–0.058)		Reference group	Reference group
Q2	≥295.22 to <545.47	412.4326	24,612	460	0.046 (0.042–0.050)		0.84 (0.74–0.95), *p* = 0.0059	0.91 (0.80–1.03), *p* = 0.1321
Q3	≥545.47 to <995.90	745.5172	24,611	365	0.037 (0.033–0.040)		0.64 (0.56–0.74), *p* = 9.46 × 10^−11^	0.76 (0.66–0.87), *p* = 9.23 × 10^−5^
Q4	≥995.90	2010.004	24,613	285	0.029 (0.025–0.032)		0.50 (0.44–0.58), *p* < 2 × 10^−16^	0.64 (0.55–0.74), *p* = 1.12 × 10^−8^
*P* for linear trend							*P* for trend < 2 × 10^−16^	*P* for trend = 4.55 × 10^−9^

^a^. Per 10,000 person-years.

**Table 3 antioxidants-12-00338-t003:** Hazard Ratios (HRs) of the associations between fCADI score and lung cancer incidence by quartiles of intake.

fCADI in Quartile	Range	Cohort	Cases	Incidence Rate (95% CI) ^a^	Age-Adjusted HR (95%CI), *p*-Value	Multi-Adjusted HR (95% CI), *p*-Value
Overall						
Q1	<−3.3440	24,613	459	0.046 (0.042–0.050)	Reference group	Reference group
Q2	≥−3.3440 to <−0.8054	24,613	417	0.042 (0.038–0.046)	0.90 (0.79–1.03), *p* = 0.127	0.85 (0.74–0.97), *p* = 0.019
Q3	≥−0.8054 to <2.4260	24,613	369	0.037 (0.033–0.041)	0.80 (0.70–0.92), *p* = 0.002	0.70 (0.60–0.82), *p* = 1.32 × 10^−5^
Q4	≥2.4260	24,612	397	0.040 (0.036–0.044)	0.89 (0.78–1.02), *p* = 0.084	0.64 (0.52–0.79), *p* = 2.78 × 10−5
*P* for linear trend					*P* for trend = 0.062	*P* for trend < 0.001

^a.^ Per 10,000 person-years.

## Data Availability

The data are contained within this article and supplementary material.

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
