# Peer review of "Association between Dietary and Supplemental Antioxidants Intake and Lung Cancer Risk: Evidence from a Cancer Screening Trial"

_antioxidants, 2023, doi:10.3390/antiox12020338_

Round 1

Reviewer 1 Report

While the manuscript is excellent as is, is there a reason why dietary supplements taken orally do not provide similar protection as those obtained via vegetable? Poor adsorptions? If high energy tumors as Lung cancers producing free radicals is protected with dietary supplements should cancers like breast (which is also described as rich in oxidative free radical) be also protected?

Author Response

Thank you for your kind review. Please find the attached response.

Reviewer 2 Report

The manuscript entitled “Association Between Dietary and Supplemental Antioxidants Intake and Lung Cancer Risk: Evidence from the Prostate, Lung, Colorectal, and Ovarian (PLCO) Cancer Screening Trial” by Jiaqi Yanget al. intends to evaluate the effect of different antioxidants on lung cancer risk reduction. The paper is interesting, well written and presented. The experimental design is appropriate. Therefore, I suggest to revise the introduction since there are not enough citations and is a little bit confusing what the authors want to tell about the role of antioxidants. For this reason, it is opinion of the reviewer, that the paper can be accepted with minor revisions.

Best regards

Author Response

(The authors gave the same response as above.)

Reviewer 3 Report

The article by Yang et al. is well written and is a very interesting work in the field of cancer screening trial.

-         The abstract, introduction and results are focused on lung cancer only while, the title included Prostate, Colorectal, and Ovarian cancer. Thus, the title is considered misleading.

-         The authors stated that “Several risk factors for lung cancer development have been identified in previous studies, such as cigarettes smoking, exposure to second-hand smoking, asbestos, and air pollution” L37 and “The results were consistent with a recent case-control study conducted in Canada that among heavy and moderate smokers, an inverse association was found between elevated dietary intakes of β-carotene, α-carotene, and lycopene in male and vitamin C in female and lung cancer risk” L311. Meanwhile, the current cigarette smokers represented only 9.3% of the study population as indicated in Table 1. How do the authors justify the impact of this small percentile on the results in context of investigation of lung cancer incidence?

Minor comments:

·      Line 111, provide the full meaning of “USDA”.

·      Line 130, provide the full meaning of “ICD-O-2”.

·      Line 162, check the structural grammar regarding the sentence “Other significant different were exhibited regarding age  ...”.

·      Reference list, some journal names were written in full while, others were abbreviated. In addition, some article titles had each word capitalized however, others were not. The authors should follow the journal style regarding this issue.

Author Response

(The authors gave the same response as above.)

Round 2

Reviewer 3 Report

I have no further comments.